# Rapid antithrombin assay from human blood plasma utilizing smartphone-based flow observation on paper chips

Ashley K. Mathews[1][¤a], Allison J. Eby[1][¤b], Jocelyn Reynolds[2], Raymond K. Wong[1], Jeong-Yeol Yoon[2]*

1 Department of Pharmacology, Perfusion Sciences Program, College of Medicine, The University of Arizona, Tucson, Arizona, United States of America, 2 Department of Biomedical Engineering, The University of Arizona, Tucson, Arizona, United States of America

¤a Current address: Stanford University Medical Center, Palo Alto, California, United States of America
¤b Current address: University of Miami Hospital, Miami, Florida, United States of America
* jyyoon@arizona.edu

## Abstract

Antithrombin (AT) is a protein that plays a critical role in regulating the coagulation cascade. Currently available methods for detecting AT levels are not portable, rapid, or simple enough to be used in urgent and clinical settings. This work presents a novel method for detecting AT levels from human blood through the smartphone-based capillary flow observation on paper chips. Assays were initially conducted with AT-spiked phosphate-buffered saline (PBS), demonstrating high linearity ($R^2 = 0.988$) between flow velocity and the logarithm of AT concentration (ng/mL), with a linear range of up to 12 ng/mL. Assays were also conducted on diluted human plasma samples, showing increasing, plateaued, and decreasing regions over the AT concentrations, respectively. A high linearity ($R^2 = 0.973$) was observed within the linear range up to 19 ng/mL. This behavior mirrors the classic Heidelberger-Kendall antibody-antigen precipitation assay, characterized by a bell-shaped curve, despite substantial differences in assay mechanisms and a concentration range that is orders of magnitude lower. To validate this similarity, we directly measured the extent of particle immunoagglutination on the paper chips using a smartphone-based fluorescence microscope and confirmed the same bell-shaped curve. These results indicate that capillary flow velocity is positively correlated with the extent of particle immunoagglutination. Using multiple human blood samples (n = 14; each subject was tested three times), the smartphone-based flow velocity AT assay was compared with ELISA, and a strong correlation was observed. The assay was essentially near-real-time, as only the first 35 frames (1.17 seconds) of data were required, provided that the antibody-conjugated particles were pre-loaded on the paper chip before the assay. It requires only a paper chip and a smartphone, and involves only one pipetting step for sample loading. This assay could one day address the critical need for

**Data availability statement:** All data are available within the article and its supplementary information.

**Funding:** One Health Initiatives, The University of Arizona.

**Competing interests:** The authors have declared that no competing interests exist.

determining AT deficiency and support clinicians in managing patients' anticoagulation status.

---

## Author summary

Antithrombin (AT) is a protein that plays a critical role in regulating the coagulation cascade. Currently available methods for detecting AT levels are not portable, rapid, and simple enough to be used in urgent and clinical settings. This work presents a novel method for detecting AT levels in human blood using smartphone-based capillary flow observation on paper chips. Experiments were conducted with AT dissolved in buffer, showing high linearity between the capillary flow velocity and AT concentration. Experiments were also repeated with diluted human blood plasma, showing the expected increasing, plateaued, and decreasing regions over AT concentration. We hypothesized that capillary flow velocity is positively correlated with the extent of particle immunoagglutination, a hypothesis confirmed by microscopic images and subsequent quantification of particle immunoagglutination directly from the paper chips. The assay was performed near real-time, required only a paper chip (pre-loaded with antibody-conjugated particles) and a smartphone, and involved a single pipetting step for sample loading. This assay could one day address the critical need to determine AT deficiency and aid clinicians in decision-making when managing their patients' anticoagulation status.

## Introduction

Antithrombin (AT) is a blood glycoprotein functioning as a serine protease inhibitor and plays an integral role in regulating the coagulation cascade. AT contains a polysaccharide-binding site, enabling its interaction with heparin. AT-heparin binding can inactivate key enzymes in the clotting cascade, including factors XIIa, XIa, and IXa in the intrinsic pathway and Xa and IIa in the common pathway ("a" denotes activated form; factor II = prothrombin; factor IIa = thrombin) (Fig 1A). Thrombin (factor IIa) is a protease that promotes clot formation by converting fibrinogen into fibrin. AT regulates clot formation by forming a 1:1 inhibitory complex with thrombin and factor Xa [1]. In the presence of heparin, AT's activity is dramatically potentiated and enhanced up to roughly 1000-fold [2]. Heparin is a polysaccharide anticoagulant that binds to AT at the polysaccharide-binding site, inducing a conformation change.

AT is clinically important to maintain hemostatic balance. AT deficiency could be hereditary or acquired over time. Hereditary AT deficiency can be further classified into two subgroups. 1) Type I is characterized by the reduced AT amount (quantitative dysfunction); 2) Type II is characterized by the reduced functional activity but no reduction in AT amount (qualitative dysfunction), which is caused by the mutations in the polysaccharide-binding site. While the prevalence of hereditary AT deficiency

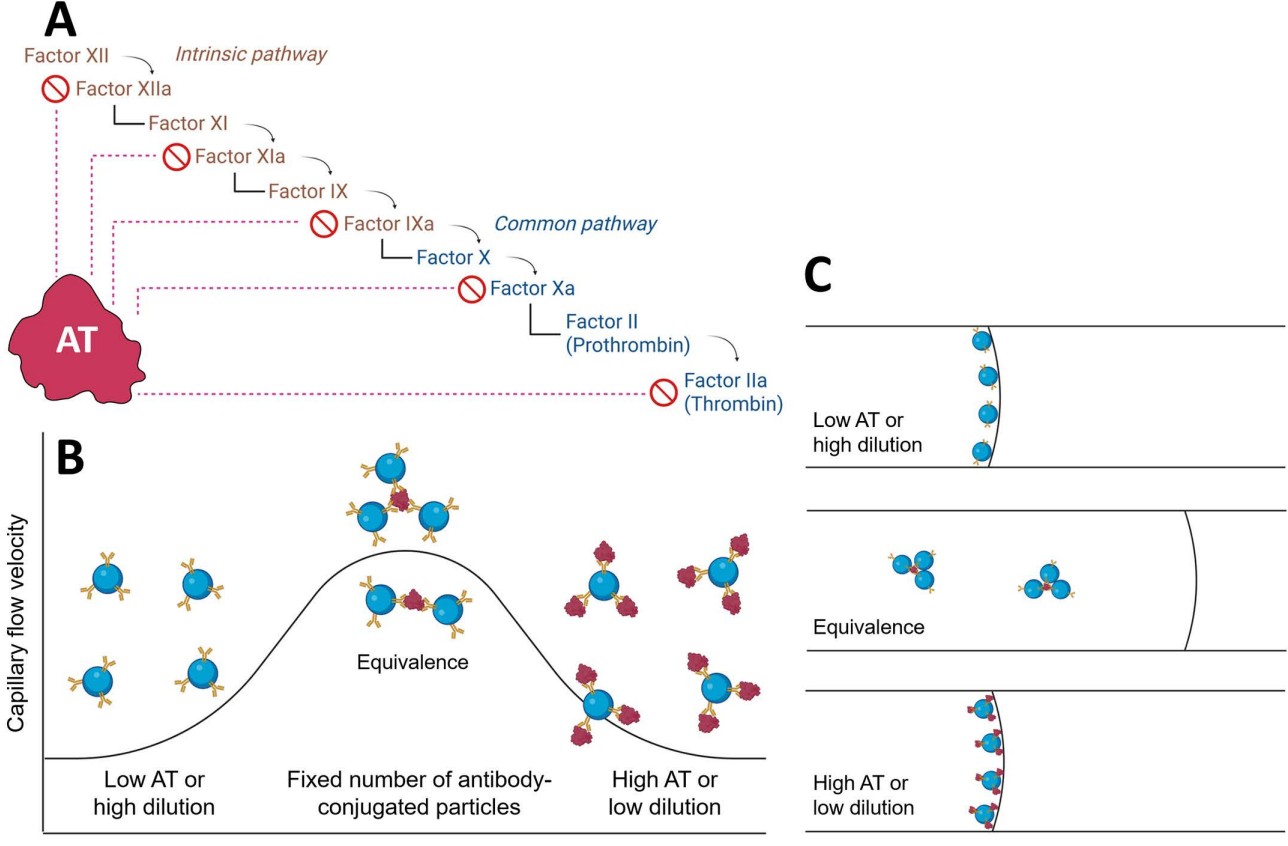

**Fig 1. The assay principle.** (A) Inhibitory role of antithrombin on the coagulation cascade. (B) A hypothesis that the capillary flow velocity measurement of particle immunoagglutination follows the Heidelberger-Kendall antibody-antigen precipitation assay. (C) Illustration of particle immunoagglutination and its retention within the paper pores, altering the surface tension at the moving front and subsequently the capillary flow velocity. Authors drew the images using BioRender.com.

ranges from 0.02% to 0.2% within the general population, these patients are at a higher risk of developing venous thromboembolism [3]. Acquired AT deficiency can be caused by several mechanisms. 1) Advanced liver disease can reduce hepatic synthesis of AT levels. 2) Nephrotic syndrome can cause AT to be lost through urinary excretion. 3) Sepsis can result in AT consumption through systemic inflammation and also coagulation activation. 4) Disseminated intravascular coagulation (DIC) can also lead to consumption of AT during widespread clotting. 5) All newborns are born with lower levels of AT compared to adults. 6) Heparin therapy can consume AT, in particular, with prolonged or high-dose treatment. 7) Major trauma and surgery can also activate the coagulation system, which can lead to AT consumption. 8) Certain malignancies can increase the activation of the coagulation cascade and AT consumption. 9) Other medications can also affect AT levels, including L-asparaginase, estrogen, and anabolic steroids. In their unique ways, these two forms of AT deficiency can reduce circulating AT, leading to an unbalanced hemostatic state and a shift towards a prothrombotic state. Ensuring adequate AT levels is critical during cardiopulmonary bypass (CPB), as heparin is administered to prevent clot formation. The target activated clotting time (ACT) for CPB is > 400 seconds when initiating and >480 seconds throughout the remainder of the case [4]. AT-deficient patients, however, are heparin-resistant, failing to secure the required ACT (i.e., failing to prolong the blood's ability to clot) even if an appropriate dose of heparin is administered. A lack of AT can imply

that heparin has little cofactor activity, leading to inadequate therapeutic anticoagulation. Prolonged CPB can also deplete the available AT in patients, leading to heparin resistance. In such cases, clinicians should administer either plasma or AT concentrate (also known as Thrombate III) [5]. AT deficiency significantly increases the risk of thromboembolic complications and clots that arise in patients who undergo CPB. These complications could lead to stroke or the death of the patient.

Therefore, there is a need for timely and efficient detection of AT deficiency, particularly in patients who undergo CPB. Identifying AT deficiency, whether hereditary or acquired, can help healthcare providers better guide their patients in making informed decisions about prophylactic anticoagulation by either administering fresh frozen plasma or Thrombate III. Although these two therapies are currently what is used for AT replacement, they have associated risks. When administering fresh frozen plasma to a patient, large volumes are required to treat AT deficiency, which can increase the risk of the patient developing transfusion-associated circulatory overload (TACO). Regarding Thrombate III, improper dosing can lead to hypersensitivity and thrombotic complications. In both instances, knowing whether a patient is deficient in AT could be clinically impactful, as it would allow clinicians to avoid treating patients prematurely and also decrease the previously mentioned risks. A portable, rapid, and simple test that accurately and rapidly quantifies AT levels remains a gap in clinical care. Such a test could immediately flag low AT levels in the operating room or any critical care setting, allowing prompt intervention for adequate anticoagulation by replenishing AT levels. The price of Thrombate III is approximately $4.66 per unit, and hospital vials range from 500 to 1000 units [6]. Clinicians administer Thrombate III to their patients when they suspect heparin-induced thrombocytopenia (HIT) or heparin resistance. A portable, rapid, and simple AT test would enable clinicians to confirm their patient's deficiency on the spot, ensuring that therapy is only given when it is truly needed. This would not only improve patient safety but also reduce unnecessary costs for patients and for the hospital's supply consumption. Such a test to detect AT would be highly instrumental in the clinical setting for effective anticoagulation patient management, especially in settings where patients who are AT-deficient undergo CPB.

Despite the clinical need, there is currently no portable method for rapidly measuring AT. The current available methods require a laboratory setting, time, and expertise. There are two broad categories of traditional AT assays: functional activity assays and immunoassays [7]. Functional activity assays are commonly used in hospitals to diagnose AT deficiency and assess AT's ability to inhibit clotting enzymes. Typically, these assays are performed by adding heparin to a patient's plasma sample and then adding a known amount of a clotting enzyme, such as factor Xa or thrombin. The mixture is incubated until AT binds it, and the added enzyme is neutralized. A chromogenic substrate is then used to measure the activity, which produces a color through enzymatic cleavage and is measured by a spectrophotometer. The less color suggests more active AT in the sample since AT must inhibit the enzyme. Immunoassay, typically an enzyme-linked immunosorbent assay (ELISA), measures AT concentration in the patient using AT-specific antibodies labeled with a color-generating enzyme-substrate pair or, sometimes, with a fluorescence dye. Unlike functional activity assays, immunoassays can be highly specific and used to validate AT levels. However, both methods have long turnaround times (at least a few hours) and require a laboratory setting to carry out the assays [8]. This can also result in relatively high costs when factoring in the reagents, equipment, and trained individuals to run the assays. Additionally, the currently available methods are not portable. Unlike POC INR (international normalized ratio) meters for warfarin monitoring, no handheld or bedside devices are available for assaying AT. There are very few papers reporting portable AT assays. Emani et al. reported functional assays on Baebies Technology's digital microfluidic platform [9], which may be too complicated (and potentially expensive) for this type of assay, and may not be as sensitive as immunoassays. Alternatively, our group has previously assessed the extent of blood coagulation by monitoring blood flow velocity through paper channels [10]. However, it did not use either an enzyme or an antibody, i.e., it did not measure AT levels.

Our laboratory has previously pioneered the use of capillary flow velocities on paper channels to quantify the extents of particle immunoagglutination [11]. Antibodies to bacteria (*Escherichia coli*) or viruses (Zika virus) were covalently conjugated to submicron-sized polystyrene particles, which were pipette-added to the paper channels

along with pathogen-containing water samples. Antigen-antibody binding caused the particles to agglutinate (immunoagglutination) within the paper's pores, leaving them behind. Such particle retention reduced the number of non-agglutinated particles at the moving front, thereby increasing the speed of capillary flow (Fig 1B). In the absence of target antigens, there is no particle retention, allowing the maximum number of non-agglutinated particles to reach the moving front and achieve the slowest capillary flow velocity. Therefore, the capillary flow velocity should increase with increasing target antigen concentration. The limits of detection (LOD) were 10 CFU/mL for *E. coli* and 20 pg/mL for Zika virus. (LOD does not apply to AT assays, since AT is always present in blood samples, and negative samples do not exist.)

However, AT detection from blood should be substantially different. Blood plasma contains a large number of diverse chemicals, and their capillary action characteristics should be significantly different from those in water samples. Our group has previously demonstrated the detection of extracellular nicotinamide phosphoribosyltransferase (eNAMPT) from human blood plasma as a biomarker for acute respiratory distress syndrome (ARDS) [12]. However, typical AT levels in plasma are 15–30 mg/dL (= 150–300 µg/mL) [13], several orders of magnitude higher than the typical eNAMPT concentrations and $10^5$-$10^7$-fold higher than the typical assay ranges for *E. coli* and Zika virus. Therefore, there is a need to redesign the assay to accommodate the clinically relevant AT levels from blood plasma.

The classic Heidelberger-Kendall curve can also be used to interpret capillary flow velocities from immunoagglutination, although the assay principles differ significantly. The Heidelberger-Kendall assay quantifies the extent of antibody-antigen binding-induced precipitations, and the resulting signal-to-antigen concentration curve is bell-shaped, characterized by three zones: 1) antibody excess, 2) equivalence, and 3) antigen excess [14]. We hypothesize that our capillary flow velocity measurement should also follow the Heidelberger-Kendall curve (Fig 1C). Given that the AT concentration ranges are quite narrow, we hypothesize that our assay curve can be shifted to the left or right by altering the dilution factor of blood plasma. More specifically, at relatively low plasma dilution factors (e.g., 0.05% or 0.01%), AT concentrations are relatively high; the curve should exhibit an inverse trend, opposite to the trend observed in [11]. There are too many AT molecules relative to the number of antibody-conjugated particles, resulting in minimal immunoagglutination. Most particles are free to diffuse to the moving front, minimizing surface tension and subsequently reducing capillary flow velocity (Fig 1C). In relatively high plasma dilution factors (e.g., 0.001%), the number of AT molecules is smaller than that of available antibodies, and the AT concentration is positively correlated to the extent of particle immunoagglutination. More immunoagglutination leads to the retention of particles in the paper's pores [15], leaving fewer particles at the moving front, thereby increasing surface tension and subsequently increasing capillary flow velocity (Fig 1C). Note that the particles are large and relatively hydrophobic, which reduces the surface tension at the liquid-air interface (= the moving front). Such hypotheses can be confirmed by imaging and counting the pixel areas of immunoagglutinated particles away from the moving front. Since our group has also pioneered the use of a smartphone-based fluorescence microscope for quantifying immunoagglutination directly from paper channels [16,17], we can employ the same method to confirm our hypotheses. The overarching aim is to explore a rapid, easy-to-use, low-cost, and portable test that can be translated into real-world clinical and field applications.

## Results

### Assay procedure

The paper chip contains four wax-printed channels, as shown in Fig 2A. Anti-AT Serpin C1 rabbit polyclonal antibody was covalently conjugated to 0.5 µm diameter, yellow-green carboxylated polystyrene particles. Particles' fluorescence was unnecessary for capillary flow velocity measurement – it was used for the later microscopic observation of particle immunoagglutination directly from the paper chips. To confirm successful antibody conjugation, we took bright-field microscopic images of antibody-conjugated particles mixed with 0.05% human blood plasma. As shown in Fig 2B, visible clumping occurred with plasma (bottom) and no clumping with deionized water (top).

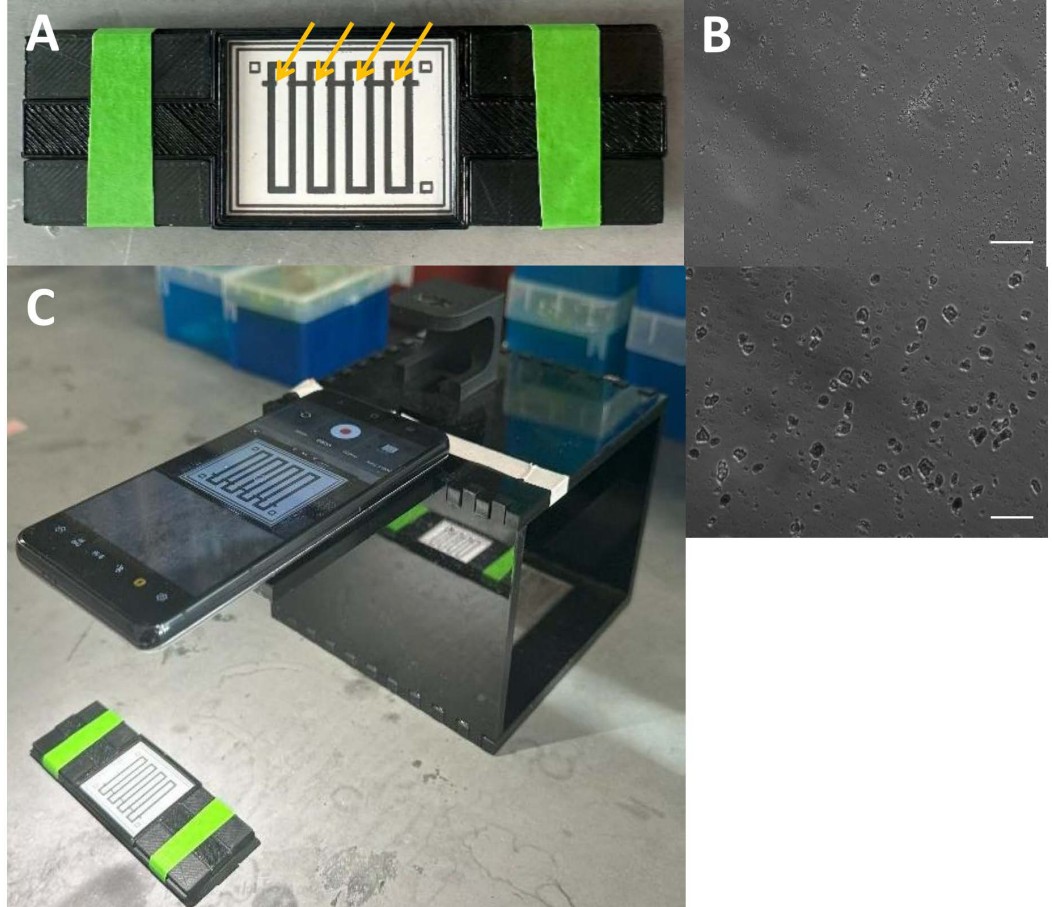

**Fig 2. Device setup.** A) A four-channel, wax-printed paper chip is placed on a custom 3D-printed holder. Arrows indicate the loading zones. B) Bright-field microscopic images of antibody-conjugated particles mixed with deionized water (top) and 0.05% human blood plasma (bottom). Scale bar = 10 µm. C) A smartphone captures a video clip of capillary action from the paper chip. Authors took the photographs.

A total of 4 µL of anti-AT-conjugated particles at 80 ng/µL were pipetted into the loading zone (inlet) of each channel and allowed to dry at room temperature before the assay. Then, 5 µL of sample (AT in PBS or diluted blood plasma) was pipetted into the loading zone of each channel, one at a time. A smartphone captured the video clips to analyze capillary flow velocities (Fig 2C).

**Capillary flow velocity assay of AT in PBS**

Capillary flow velocities were collected over the following time frames: 0–25, 0–30, 0–35, and 0–40 (note: videos were collected at 30 frames per second), using AT solutions dissolved in PBS. The average flow velocities with six different AT concentrations were collected, and each data set was analyzed with ANOVA to calculate p-values. The resulting p-values were 0.254 for 0–25 frames, 0.176 for 0–30 frames, 0.038 for 0–35 frames, and 0.247 for 0–40 frames, indicating that the 0–35 frames represented the best antibody-antigen binding and subsequent particle immunoagglutination. The average flow velocities for different AT concentrations (n = 8 for each concentration) are shown in Fig 3A, and the corresponding standard errors are shown in Fig 3B. The standard error decreased as the time interval increased, presumably because including longer time-frame data stabilized the data fluctuations. However, starting from 0-40, the standard error began

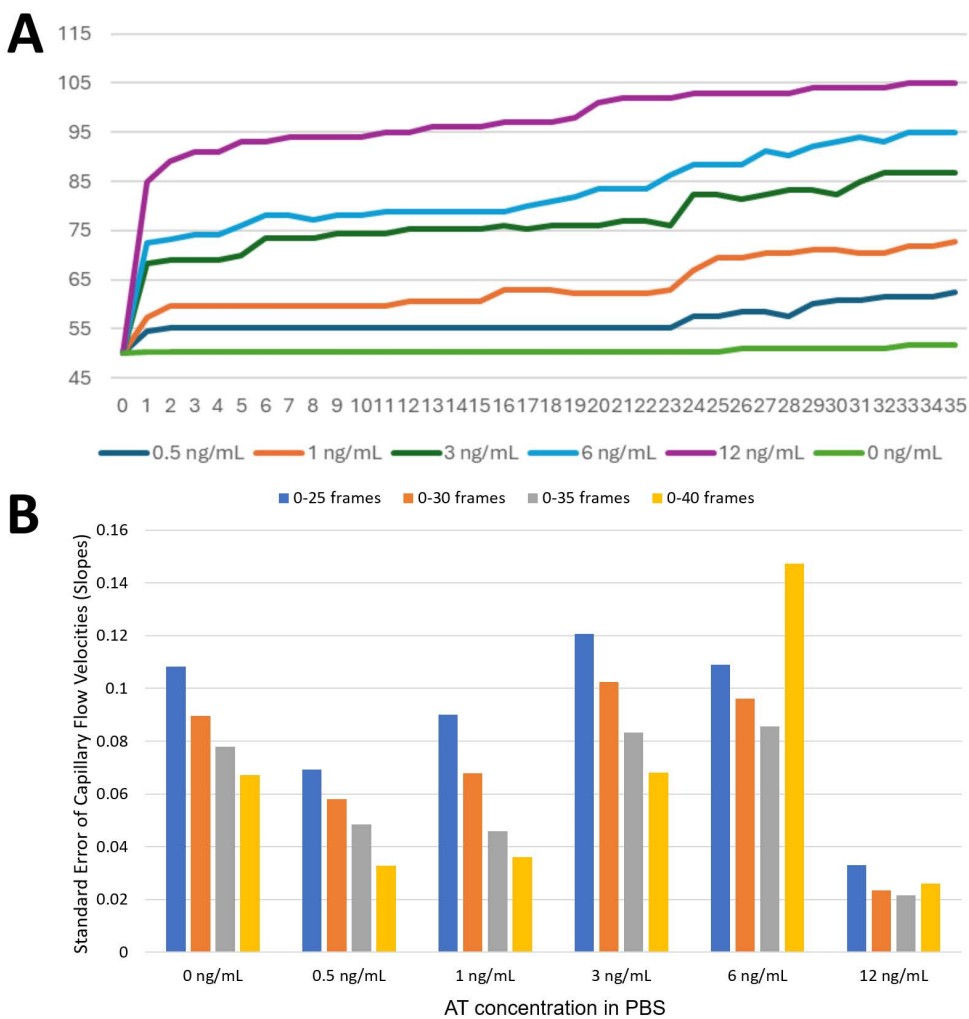

**Fig 3. Time frame optimization for evaluating capillary flow velocities with AT solutions in PBS.** A) Average capillary flow profiles (n = 8). B) Standard errors of capillary flow velocities (= slopes of capillary flow profiles) (n = 8) for the first 0-25, 0-30, 0-35, and 0-40 frames. Videos were captured at 30 frames per second. Authors drew the charts using Microsoft Excel.

to rise, likely due to anomalies in the paper channels (residual wax, dust contamination, non-homogeneous pores, etc.). Therefore, capillary flow velocities were collected from the first 35 time frames (0–35), which corresponded to the first 1.17 seconds of the flow.

When the average flow velocities were plotted against AT concentration, the curve exhibited the expected logarithmic increase (Fig 4A), similar to the left side of the bell-shaped curve in Fig 1B. When the curve was plotted against the logarithm of AT concentration (data at 0 ng/mL were excluded in the logarithmic plot), a good linear relationship was observed, with an R² value of 0.988 (Fig 4B). The slope (0.136) indicates a semi-logarithmic relationship between capillary flow velocity and AT concentration, as previously demonstrated with the *E. coli* assays [12]. In fact, particle immunoagglutination assays have previously been shown to exhibit a similar semi-logarithmic relationship when monitored by light-scattering intensity [18] and by particle counting [16]. The y-intercept (0.575) represents the flow velocity in the absence of AT, i.e., the flow velocity with the particles not immunoagglutinated. Given that there was no plateau or decrease in the flow velocity, it

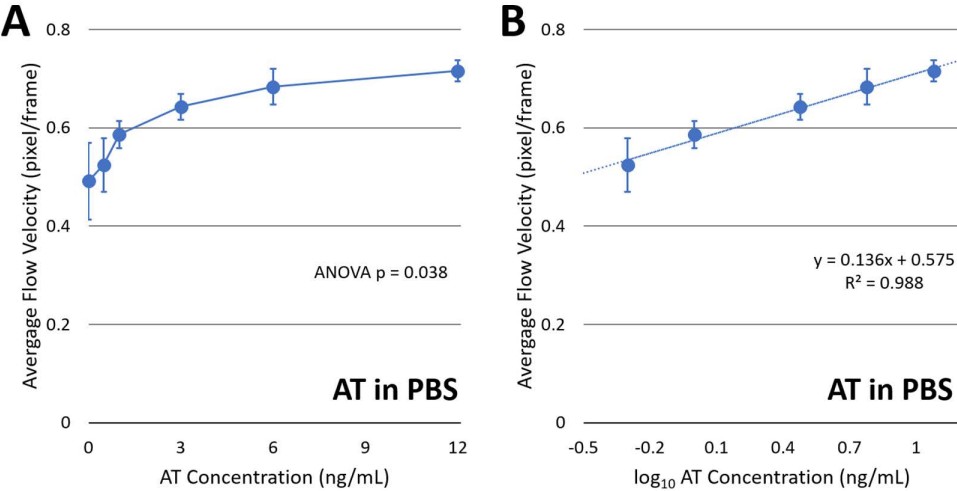

**Fig 4. Flow velocity assays with AT in PBS.** A) Average flow velocities (pixel/frame) plotted against AT concentration in ng/mL (n = 8). B) The same plotted against logarithmic AT concentration. Error bars represent standard errors. Authors drew the charts using Microsoft Excel.

can be assured that there was no oversaturation region in the AT concentrations tested in this experiment. The standard error bar is largest at 0 ng/mL, potentially indicating the need for AT to stabilize the anti-AT-conjugated particles.

## Endogenous AT levels in human blood samples

Human blood samples were collected from volunteers in accordance with an IRB-approved protocol (STUDY00004578), and their endogenous AT levels were determined using a commercial AT ELISA kit (SimpleStep Human Antithrombin ELISA kit, catalog number ab222507, Abcam Limited, Cambridge, UK). The total number of human subjects was 14. The first two subjects' samples, labeled A and B, were used to construct a standard curve. The remaining 12 subjects' samples, labeled 1–12, were used to correlate the flow-velocity assay with ELISA, as discussed later.

Fig 5 shows the ELISA assay results. Since the kit detects AT level in the ng/mL range, we diluted the samples to 0.01%. The standard curve is shown in A), using a series of AT solutions in PBS, ranging up to 50 ng/mL. With 0.01% plasma, the normal range corresponds to 15–30 ng/mL. The obtained concentrations were converted to 0.001% plasma concentrations to compare them with those from flow velocity assays adequately. The results are shown in B), showing that the majority of samples fall in the normal range of 1.5-3.0 ng/mL, or at least very close to it.

Subjects A and B, which were used to construct standard curves, showed the AT levels of 1.59 ng/mL (subject A) and 2.30 ng/mL (subject B) from 0.001% plasma, both within normal ranges (1.5-3 ng/mL in 0.001% plasma or 15–30 mg/dL in undiluted plasma) [13]. The undiluted sample from subject A was serially diluted at 10-fold increments to 0.001% and 0.01% concentrations, corresponding to expected endogenous AT levels of 1.59 and 15.9 ng/mL, respectively. We then spiked additional AT to these dilutions, increasing the concentrations to 1.59, 2.09, 2.59 ng/mL and 15.9, 16.4, 16.9, 18.9, 27.9 ng/mL, respectively. Normal range of 15–30 mg/dL in undiluted plasma corresponds to 1.5-3 ng/mL in 0.001% plasma and 15–30 ng/mL in 0.01% plasma.

## Capillary flow velocity assay of at in human blood plasma

Building on the initial calibration with PBS samples, we proceeded to test the assay's ability to function in a complex matrix, diluted plasma. Fig 6A shows the results with diluted plasma containing varying amounts of spiked AT. At low AT concentrations, it shows the expected increasing trend similar to Fig 4A, corresponding to the left side of the bell-shaped

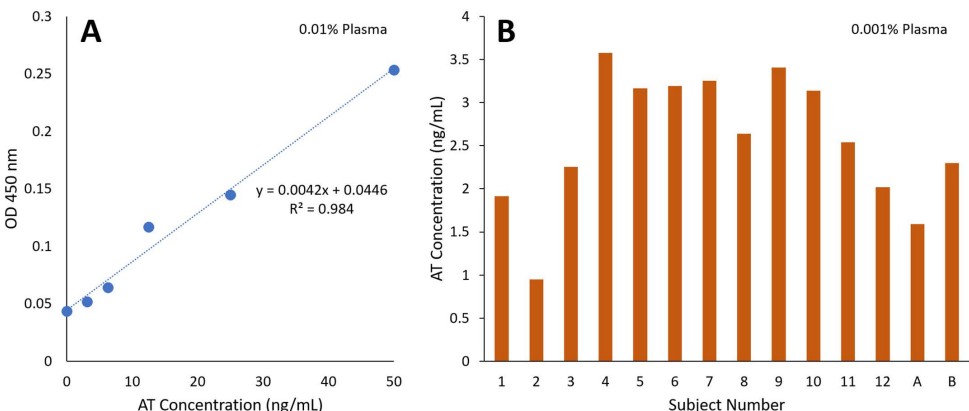

**Fig 5. ELISA assays of human plasma samples.** A) A standard curve using a series of AT solutions in PBS. Average of three measurements for each data point. B) AT concentrations in 0.001% plasma from 13 human subjects. Averages of three measurements for each subject. AT concentrations were calculated using the standard curve equation shown in A). Authors drew the charts using Microsoft Excel.

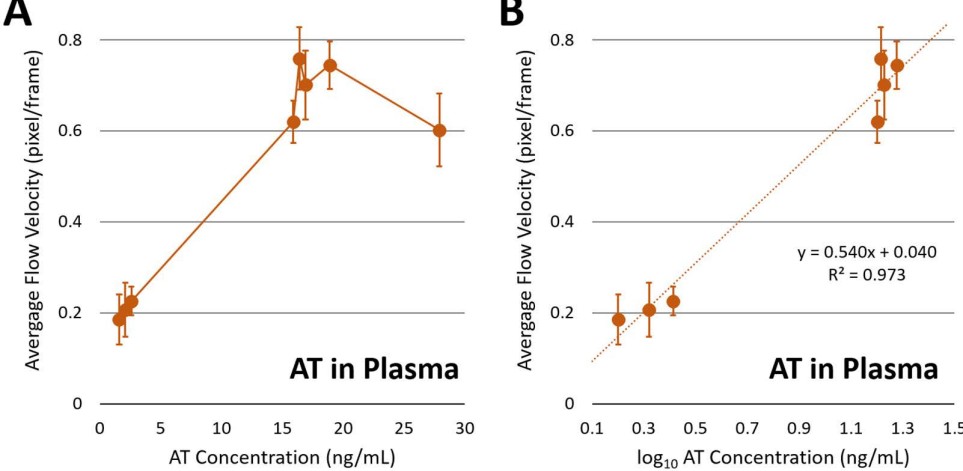

**Fig 6. Flow velocity assays with AT-spiked human blood plasma.** A) A human plasma sample with the endogenous AT level of 1.59 ng/mL was diluted to 0.001% and spiked with 0, 0.5, and 1 ng/mL AT, yielding 1.59, 2.09, and 2.59 ng/mL. It was also diluted to 0.01% and spiked with 0, 0.5, 1, 3, and 12 ng/mL AT, yielding 15.9, 16.4, 16.9, 18.9, and 27.9 ng/mL. Average flow velocities (pixel/frame) were plotted against AT concentration (n = 7 or 8). Error bars represent standard errors. B) The same plotted against logarithmic AT concentration to generate the standard curve equation. Authors drew the charts using Microsoft Excel.

curve (Fig 1B). At high AT concentrations, capillary flow velocities increased substantially, leading to a much steeper increase. For the given linear range, the curve was plotted against the logarithm of AT concentration, and a good linear relationship was observed, with a much higher slope (0.540 versus 0.136 in PBS) and an $R^2$ value of 0.973 (Fig 6B). Apparently, the presence of human blood plasma amplified the extent of particle immunoagglutination, potentially through minimizing non-specific immunoagglutination. However, this presumption could not be confirmed, as fluorescence staining and subsequent microscopic imaging of numerous plasma components are challenging on a paper substrate (because of their small sizes) [19].

Flow velocities eventually plateaued at around 19 ng/mL and decreased with higher concentrations, corresponding to the equivalence and antigen-excess regions of the bell-shaped curve shown in Fig 1B. When AT concentrations are too high, excess antigen saturates the available antibodies, inhibiting effective crosslinking of particles and reducing aggregation. Rather than being inconsistent, these results express that too much AT would decrease the flow velocity slope. It became evident that endogenous AT in the plasma samples contributed significantly to the oversaturation effect, masking the dynamic range of our assay's detection system.

Overall, the capillary flow velocity assay of AT in diluted plasma closely matched the classic Heidelberger-Kendall antibody-antigen precipitation assay, despite differences in detection methodology. This result provides indirect evidence of a positive correlation between particle immunoagglutination and capillary flow velocity.

### Particle counting assay of AT in human whole blood

To confirm whether the capillary flow velocity is truly correlated to the extent of particle immunoagglutination, we captured fluorescence microscopic images of the immunoagglutinated particles directly from the same paper chips. A smartphone-based fluorescence microscope was used (Fig 7A), as previously demonstrated by our group [16,17]. The pixel counts of immunoagglutinated particles (after background noise removal and separation from non-agglutinated particles) were evaluated across multiple areas of the paper channel, away from the moving front (i.e., the particles left behind). Only the pixel areas were assessed, not the fluorescence intensities, as quantifying fluorescence intensity on a paper substrate is proven to be substantially challenging [19].

Fig 7B shows representative smartphone-based fluorescence microscopy images of particle immunoagglutination from the paper chips, with raw (top) and processed (bottom) images. Samples from subject B were used for the assay, with varying spiked AT concentrations in diluted whole blood. Fig 7C shows the plot of pixel counts (corresponding to the extent of particle immunoagglutination) against spiked AT concentration, clearly showing the expected bell-shaped curve. These findings clearly indicate that particle immunoagglutination is positively correlated with capillary flow velocity measurements.

The equivalence zone can be observed at approximately 1 ng/mL in 0.001% whole blood, which is significantly lower than the 16–19 ng/mL range typically observed in capillary flow velocity measurements. However, the capillary flow velocity assay remains superior, as it is much easier to operate and exhibits a linearly increasing response with 0.001% plasma samples. (Particle counting assay requires multiple microscopic images from paper chips, a dark chamber, robust lighting conditions, and manual focusing of submicron-size particles.)

### Correlation between capillary flow velocity assay and ELISA using multiple human plasma samples

To validate against the laboratory standard, we conducted capillary flow velocity assays on human plasma samples from multiple human volunteers. Human whole blood samples were centrifuged to obtain plasma, which was then diluted to 0.001% in PBS. Capillary flow velocity assays were conducted in an identical manner described above. Three independent assays were performed for each human subject's sample. For each assay, flow distances were measured at four lines and averaged to ensure accurate readings. Therefore, the total dimension of data was $3 \times 4 = 12$ for each subject. The flow velocities from the first 35 time frames were collected. The standard curve equation obtained from spiked human plasma samples, $y = 0.540x + 0.040$ (Fig 6B), was used to calculate AT concentrations.

Fig 8A shows the correlation between the flow velocity and ELISA-measured AT concentrations in 0.001% plasma. The slope is 1.06 and is quite close to 1, indicating the strong correlation between the two assays. Several outliers are evident (circled), which can be attributed to 1) subject-to-subject variation in plasma composition affecting the capillary flow, 2) potential errors in ELISA (incomplete rinsing and/or incubation), 3) potential defects in paper chips (residual wax blocking capillary flow), etc. These factors will be revisited in the Discussion.

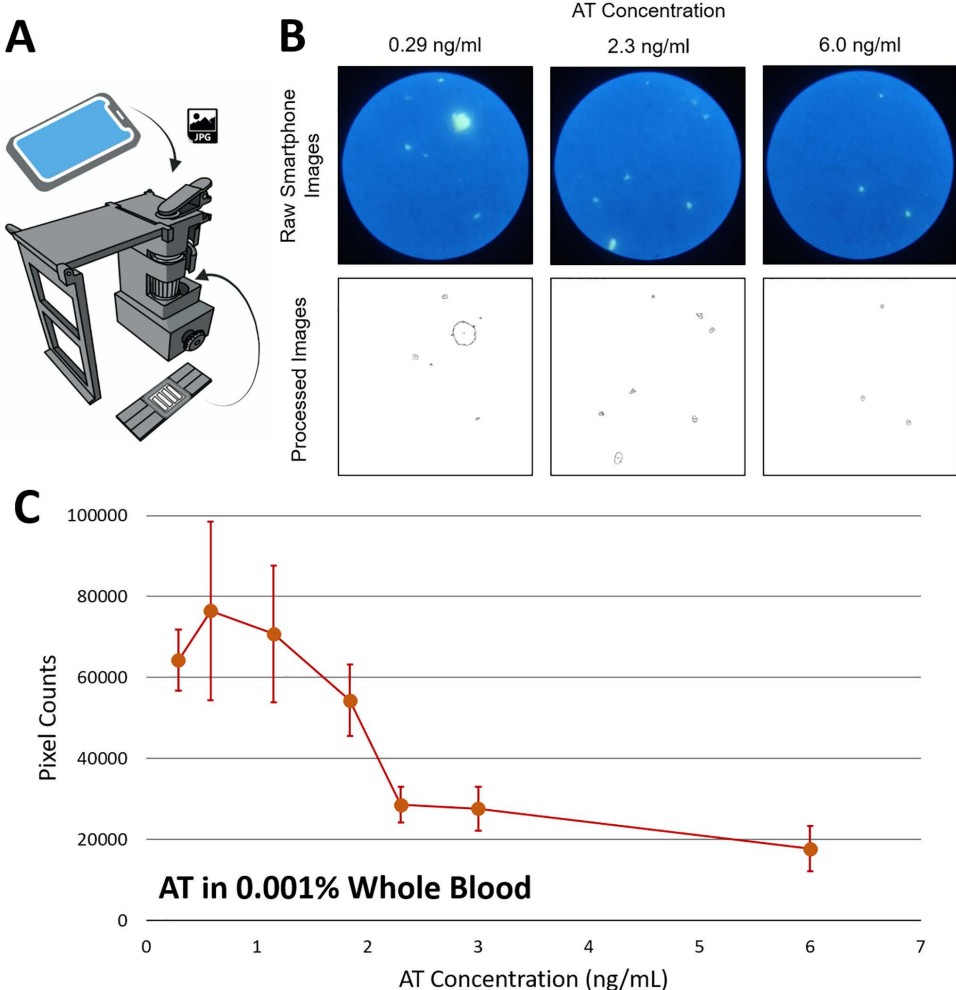

**Fig 7. Particle counting assays with AT-spiked human whole blood.** A) Device schematics and assay procedure. B) Raw and processed images of particle immunoagglutination from the paper chips. C) The pixel counts (representing the extent of particle immunoagglutination) are plotted against the AT concentration (n = 8). 0.001% whole blood was used. Error bars represent standard errors. Authors drew the smartphone microscope design using SolidWorks, photographed the microscopic images, and drew the charts using Microsoft Excel.

We repeated the flow-velocity assays using the higher antibody-conjugated particle concentration on the same set of human plasma samples. The flow velocities were noticeably higher at the higher particle concentration, so we introduced a universal correction factor of 1.5 to calculate AT concentrations correctly. Fig 8B shows the result. It shows improved correlation with fewer outliers, though the results are not substantially different from Fig 8A.

## Discussion

This work aims to provide a novel proof-of-principle AT assay that leverages changes in capillary flow velocity arising from particle immunoaggregation. Capillary flow profiles were collected from distance (in pixels) versus time (in frame numbers) plots, and the first 0–35 frames (1.17 seconds) showed a pronounced difference between AT concentrations. AT-spiked PBS samples exhibited a clear logarithmic-linear response up to 12 ng/mL AT, indicating the assay's high sensitivity and linearity. Diluted plasma samples (0.01% and 0.001%) were also tested with varied amounts of spiked AT, showing the

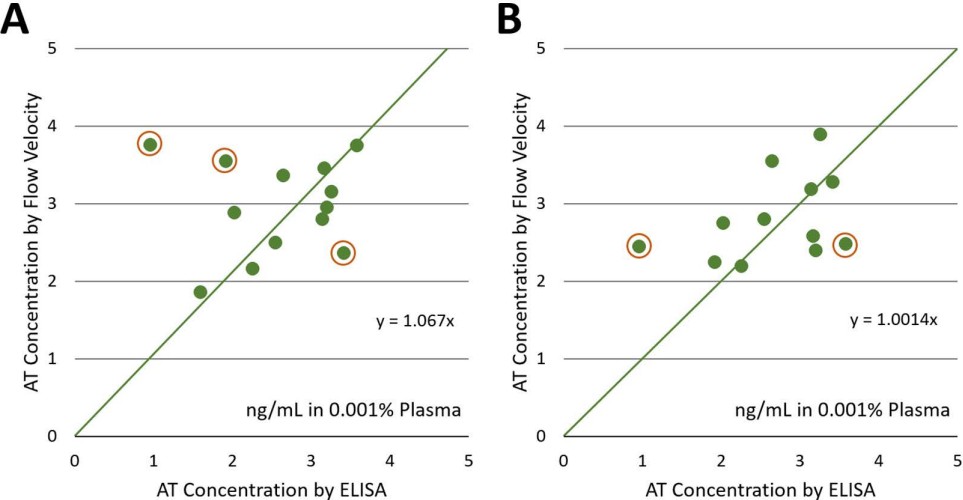

**Fig 8. Correlation between flow velocity- versus ELISA-measured AT concentrations.** A) AT concentrations were calculated using the standard curve shown in Fig 5B: y = 0.540x + 0.040. Averages from three measurements for each human subject sample. They are plotted against the ELISA concentrations shown in Fig 7. Samples tested: 1-12 and A. B) Higher concentrations of antibody-conjugated particles were used, and the flow velocities were measured (n = 3 per subject). The calculated AT concentrations were divided by a universal correction factor of 1.5. Samples tested: 1-12. For both A) and B): Averages from three different experiments for each sample, and four line distances per each experiment. Outliers are circled in both plots. Authors drew the charts using Microsoft Excel.

expected linear increase (antibody excess) and saturated/decreased regions (equivalence/antigen excess), respectively. While higher dilutions of 0.05% and 0.1% were also tested, their results were not reproducible, and the linear ranges were very narrow, corroborating that 0.001% dilution is optimal for this assay.

These results also mirrored the classic Heidelberger-Kendall antibody-antigen precipitation assay, represented by a bell-shaped curve with three zones: antibody excess, equivalence, and antigen excess. The capillary flow velocity was assumed to be positively correlated with the extent of particle immunoagglutination, despite the mechanisms differing and the concentration ranges being substantially different. To confirm this similarity, we directly measured the extent of particle immunoagglutination on the paper chips using a smartphone-based fluorescence microscope and observed a similar bell-shaped curve.

A logarithmic-linear relationship was observed in both AT-spiked PBS and diluted plasma samples, with very high $R^2$ values of 0.988 and 0.973, respectively. The slopes of standard curves, on the other hand, were quite different: 0.136 in PBS and 0.540 in diluted plasma. The most abundant plasma protein is albumin, which has been widely used in immuno-assays to passivate surfaces [8]. Such passivation minimizes non-specific antibody-antigen binding. In fact, the y-intercept of the standard curve is quite low, 0.040 with diluted plasma, compared to 0.575 with PBS. Such passivation of non-specific antibody-antigen binding maximizes the participation of antibody-conjugated particles in immunoaggluti-nation, leading to high sensitivity and, consequently, a steeper slope in the standard curve. Albumin can also stabilize the antibody-conjugated particles, minimizing non-specific particle aggregation.

The flow velocity assay was compared with a commercial ELISA kit using multiple human subject samples. It showed linear correlation between the two assays, although several outliers were observed. Such outliers can be attributed to 1) variations in plasma samples, 2) errors in ELISA, and 3) defects in the flow velocity assay. Firstly, the composition of plasma samples should vary from subject to subject, including albumin and fibrinogen. Albumin should play an important role in passivating non-specific immunoagglutination, which should affect the flow velocity assay either positively or negatively. Fibrinogen is responsible for blood coagulation, which should directly impact

the capillary flow velocity. While the dilution of 0.001% should decrease their concentrations substantially, e.g., 45 mg/mL albumin in undiluted plasma is equivalent to 450 ng/mL in 0.001% plasma, they are still sufficiently higher than the detection limits of capillary flow assays (pg/mL scale) [11,12] and can make substantial impacts on particle immunoagglutination and flow velocity assay. Secondly, ELISA is not perfect and prone to user error, such as inconsistent pipetting, incomplete rinsing, and suboptimal incubation. One human subject sample (#2) showed a very low AT concentration of 0.95 ng/mL, substantially lower than the normal range of 1.5-3.0 ng/mL in 0.001% plasma (= 15–30 mg/dL in undiluted plasma). However, the flow velocity assay reported much higher AT concentrations in both Fig 8A and 8B, creating the most obvious outlier at the far left, potentially indicating an error in the ELISA. Thirdly, the wax-printing process for fabricating paper chips may also be imperfect, as residual wax can remain in the main channel and subsequently block capillary action [16]. In such a case, the capillary flow velocity should be lower than expected, and the AT concentration should be underestimated. The outliers at the bottom-right in Fig 8A and 8B are potential examples.

Once the sample was loaded onto the paper chip, it took less than 2 minutes to reach the end of the channel. However, there is no need to wait until 2 minutes, since only the first 0–35 frames (1.17 seconds) of data were needed. As long as the antibody-conjugated particles are pre-loaded, dried, stored in a refrigerator, and provided to the experimenter, the assay can be conducted in near-real time (within seconds). The majority of assay time will be spent on plasma dilution. It only requires a paper chip (pre-loaded with antibody particles) and a smartphone, and involves only one pipetting step for sample loading. In other words, it requires neither expensive laboratory equipment nor extensive expertise to carry out the assay.

This work can be adapted to quantify factors involved in blood coagulation, such as prothrombin time (PT), activated partial thromboplastin time (aPTT), thrombin time (TT), fibrinogen assay, von Willebrand factor (VWF) assay, anti-Xa assay, and lupus anticoagulant testing, among others [20].

While the assay demonstrated the desired trends and provided proof of concept, several setbacks remain. Firstly, the assay's sensitivity was too high, requiring plasma dilution to 0.001%, necessitating an additional sample preparation step; however, similar limitations persist in other established assay methods as well. The concentration of antibody-conjugated particles can be increased substantially to accommodate low dilutions, e.g., 0.1% or 1%. (The increase in antibody-conjugated particles' concentration does not need to be several orders of magnitude, as the assay is linearly proportional to the logarithmic AT concentrations.) However, such an increase is likely to compromise sensitivity.

At this stage, diluting to 0.01% or 0.001% requires several steps of manual pipetting. While automated blood or plasma dilution is outside the scope of this proof-of-concept demonstration, numerous methods are currently available, including commercial sample dilution devices such as Dilumat® from bioMérieux [21] and microfluidic serial dilution devices [22].

Secondly, the flow velocity data can be influenced by the characteristics of the nitrocellulose paper. Different types (and even batches) of nitrocellulose paper significantly altered capillary action, so we had to use the same type and batch to ensure assay reproducibility. Environmental factors, particularly relative humidity (RH), also significantly impacted the results. While we collected our data with RH < 40%, the results varied significantly when RH increased over 50%. Thirdly, there are limitations due to manual pipetting. Automated sample loading that delivers precise volumes could substantially enhance the assay's user-friendliness.

Thirdly, the assays were not tested in the presence of other medications, as the patients in intensive care settings often receive other drugs that may affect the extent of blood coagulation. The impact of these additional medications should be negligible, as the antibody to AT is generally considered highly specific and less likely to bind them. In addition, the capillary flow velocity change should primarily result from surface tension changes induced by particle immunoagglutination, as demonstrated in our previous work on the SARS-CoV-2 assay in human saliva [17]. Nonetheless, it will be worthwhile to include such results in the future, especially from patients' samples in intensive care settings, to investigate the effects of various medications.

Fourthly, this assay aims to replace quantitative AT assays but would not replace AT activity assays. Such AT activity assays are required to diagnose qualitative defects in AT function, which is not the aim of this study.

To apply this preliminary proof-of-principle for a clinical setting, several further studies and modifications may be necessary. While smartphone-to-smartphone variations should be minimal, since the method does not "quantify" optical density but rather monitors flow motion, the edge-detection parameter may need to be adjusted slightly depending on ambient lighting conditions. In a situation where smartphone use is forbidden, a smartphone can be replaced with a Raspberry Pi and a Pi camera, which can also be interfaced with cloud-based analysis. An automated sampling loading system can also make the system more user-friendly. An open-source automated pipette method can be used alongside the proposed method, including https://github.com/ac-rad/digital-pipette or the one demonstrated by our group [23].

## Conclusion

The work aims to fill a critical gap in the clinical need for a novel, portable, and rapid test for AT levels. It uses capillary flow velocity on paper chips to measure particle immunoagglutination and detect AT levels from human plasma. It followed the classic Heidelberger-Kendall antibody-antigen precipitation assay, with a characteristic bell-shaped curve, indicating that capillary flow velocity is positively correlated with the extent of particle immunoagglutination at lower concentrations. The assay achieved satisfactory sensitivity with a much quicker turnaround time than currently available methods for quantifying AT. The assay method is simple, low-cost, rapid, and user-friendly. However, substantial refinements are still needed for our assay to be effective in the clinical setting. This assay could one day address the critical need to determine AT deficiency and aid clinicians in their decision-making when managing anticoagulation in their patients.

## Materials and methods

### Fabrication of paper chips

A typical paper-based microfluidic chip (hereafter, a paper chip) requires hydrophobic barriers to define its channel boundaries and flow regions. Unisart(R) nitrocellulose membrane CN95 (Sartorius AG, Göttingen, Germany) was chosen as the substrate for paper chips, as previous work has shown its capability to minimize non-specific fouling of particles and proteins [11,16]. This membrane has a capillary speed of 65–115 seconds/40 mm, a thickness of 240–270 μm, and an approximate pore size of 15 μm. The four-channel layout was created by SolidWorks (Fig 2A) and printed on the nitrocellulose membrane using a commercial wax printer (ColorQube 9550; Xerox Corporation, Norwalk, CT, USA) [24]. Two small squares at the top of the chip serve as visual indicators to mark the side on which the sample is placed. Each channel consists of a 0.3 x 0.3 cm sample-loading zone. The horizontal line that separates the loading zone and the main channel is where the antibody-conjugated particle suspension is pre-loaded and dried before loading the AT/blood samples. Capillary action drives the sample fluid downward through each channel. A single square is located in the bottom-right corner to indicate the bottom location and distinguish it from the top. After wax printing, the piece was cut out and heated on a hot plate for 3 minutes at 120 °C with a metal weight and a piece of foil placed on top of the paper chip. This process melted the wax throughout the entire depth of the paper substrate, creating a uniform and consistent hydrophobic barrier.

### Antibody conjugated particles

Since AT antibodies are not naturally present in humans, an anti-AT SerpinC1 rabbit polyclonal antibody was used and conjugated to 0.5 μm diameter, yellow-green carboxylated polystyrene particles (Magsphere, Inc., Pasadena, CA, USA). This antibody binds the C-terminus of AT. Since there are two binding sites on AT, the C-terminus does not interfere with the heparin-binding site. Antibodies were covalently conjugated to the particles using 1-ethyl-3-(3-dimethylaminopropyl)carbodiimide (EDAC), as reported in [11] and [16]. The resulting antibody-particle suspension was further diluted 1:100, and a miniature spectrophotometer (Flame; Ocean Optics, Orlando, FL, USA) was used to quantify the particle concentration using an

established standard curve. Through further dilutions, the concentration was set to 80 ng/μL. When verifying the assay by ELISA, a higher particle concentration of 100 ng/μL was also used, and the results are shown in Fig 8B. Aliquots were made to minimize the destabilization of antibody-particles. Before running the experiments, these particles were sonicated for 10 minutes to break up unwanted aggregation and achieve an even particle distribution within the solution.

Before conducting the main experiments, we obtained bright-field microscopic images (Eclipse TS100; Nikon, Tokyo, Japan) of antibody-conjugated particles mixed with 0.05% human blood plasma, which should contain sufficient AT. Fig 2B shows evident clumping with plasma (right) and no clumping with deionized water (left).

**Capillary flow velocity measurements**

4 μL of anti-AT-conjugated particles at 80 ng/μL were pipette-added to the loading zone (inlet) of each channel and dried at room temperature for 10 minutes. In any instance where the particle-loaded paper chips could not be used within 1 hour, they were stored in sealed containers and refrigerated at 4 °C. The refrigerated particle-loaded paper chips were allowed to reach room temperature for 10 minutes before use.

For each sample run, a paper chip was placed in a custom 3D-printed holder (Fig 2A) to maintain a fixed position relative to the smartphone camera (Galaxy S21, Samsung, Suwon, Republic of Korea). The smartphone's default video app was used with standard high-speed video recording settings, flash activation, and fixed exposure. Once the paper chip was secure and the camera was in position for recording, video recording started for the sample runs (Fig 2C). 5 μL of sample (AT in PBS, diluted human blood plasma, or diluted human whole blood) was pipette-added to the loading zone of each channel, one at a time, ensuring full coverage of the loading zones. Video clips were recorded to evaluate capillary flow velocity in the samples through paper channels for approximately 2 minutes. All samples were observed to have travelled the entire distance of each channel of the paper chip in less than 2 minutes. Time-frame optimization data with AT in PBS can be found in the S1 File. Raw flow profiles of AT-spiked blood plasma used to establish the standard curve are available in the S2 File. The processed flow velocities of AT in PBS and AT-spiked plasma used to establish the standard curves are provided in the S3 File.

**Particle counting measurements**

As an alternative to capillary flow measurements, the extent of particle immunoagglutination was directly quantified from the same paper chips using a smartphone-based fluorescence microscope, following a protocol previously developed by our group [16,17]. The design and image of the smartphone-based fluorescence microscope are also found in [17]. Three images were captured from the paper channels for a single assay, away from the moving front. A separate blue LED illuminated the yellow-green particles on the paper chips, and a green bandpass filter was attached to the smartphone's camera to capture green fluorescence. Identical particles (yellow-green fluorescent polystyrene particles) were used for both assays (capillary flow velocity and particle counting), although the particles' fluorescence was unnecessary for capillary flow measurements. The green channel images were isolated from the raw images. Pixels with an intensity below a set threshold (100 out of 255) were eliminated to remove background noise. These thresholded images were then binarized. Particles were identified from these binarized images, and the histogram of the particle sizes (in pixel numbers) was obtained. Particles with sizes smaller than a set threshold (50) were considered non-agglutinated and removed from the count [16]. The pixel counts of all remaining particles were summed across the three images, representing the extent of particle immunoagglutination remaining within the paper pores. The processed particle counting data can be found in the S4 File.

**Data analysis on Google Colab**

The video recordings of capillary flow were automatically uploaded to Google Drive and analyzed using a custom Python script running on Google Colab. The edge-detection algorithm (using the OpenCV library) differentiated the wet/dry boundaries of the sample as it traveled through the channels, frame by frame. From this code, the distance-versus-time

frame profiles were compiled for each channel. The raw experimental data were checked for any obvious errors (e.g., zero or maximum distance at the middle of the flow) that were manually eliminated. In addition, the "inflection point" where the flow started was manually identified, and the distance-frame profile was shifted so that the plot begins at zero-time frame.

The smartphone-based microscopic images were processed similarly, i.e., using Google Drive and a custom Python script running on Google Colab. Codes and examples can be found at [16].

### AT and blood samples tested

Initially, AT solutions were dissolved in phosphate-buffered saline (PBS). AT concentrations of 0, 0.5, 1, 3, 6, and 12 ng/mL were used. The rationale is as follows. The normal AT levels in human plasma are 15–30 mg/dL (= 150–300 μg/mL). In 0.001% plasma ($10^{-5}$ dilution), they are equivalent to 1.5-3 ng/mL [13]. With 0.01% plasma, they are comparable to 15–30 ng/mL.

Human whole blood and subsequent human plasma samples were acquired from human volunteers. The University of Arizona's Institutional Review Board (IRB) approved the protocol, designated as STUDY00004578. AT was also spiked into the diluted human plasma and diluted human whole blood. Two different plasma dilutions were tested: 0.01% and 0.001%. One whole-blood dilution was used: 0.001%. Serial dilutions were made at 10-fold increments to reach 0.01% or 0.001%.

AT levels in human blood plasma and whole blood were quantified using a commercial AT ELISA kit, SimpleStep Human Antithrombin ELISA kit (catalog number ab222507; Abcam Limited, Cambridge, UK). This kit detects AT levels in the ng/mL range, with the lowest detectable concentration of 0.33 ng/mL. The standard curve and the calculation of endogenous AT levels can be found in Fig 5. Raw data for all ELISA tests are available in the S5 File.

### Correlation between capillary flow velocity assay and ELISA using multiple human plasma samples

Flow velocity assays were repeated with multiple human plasma samples (diluted to 0.001%) using the identical method described above. The standard curve equation of AT-spiked diluted human plasma samples (Fig 6) was used ($y = 0.540x + 0.040$) to calculate AT concentration. For each experiment, flow distances were measured at four lines and averaged to ensure accurate readings. Three plasma samples were tested for each subject, yielding a total of 12 data dimensions per subject. These flow-velocity assay results were plotted against ELISA concentrations, using two different antibody-conjugated particle concentrations (Fig 8). Raw flow profiles for all correlation assays are in the S6 and S7 Files, and the corresponding processed flow velocities are in the S8 File.

### Statistical Analysis

The capillary flow velocities were plotted against the AT concentration, and linearity can be observed when the capillary flow velocities were plotted against the logarithmic AT concentration. The linear trend lines were identified, and the coefficient of determination ($R^2$) was calculated in Microsoft Excel. A comparative correlation analysis was also conducted using one-way ANOVA across different AT concentrations, using Microsoft Excel.

### Supporting information

**S1 File. Optimization of the time interval for evaluating average capillary flow velocities.** AT solutions in PBS with 0, 0.5, 1, 3, 6, and 12 ng/mL were used. Assays were repeated over eight different channels (n = 8).
(XLSX)

**S2 File. Raw capillary flow profiles with AT-spiked human plasma samples (0.001% and 0.01%).** Assays were repeated over seven to eight different channels (n = 7 or 8).
(XLSX)

**S3 File. Processed flow velocity data with AT in PBS, 0.001% plasma, and 0.01% plasma.**
(XLSX)

**S4 File. Processed particle counting data with AT in 0.001% whole blood.**
(XLSX)

**S5 File. ELISA of human blood samples.**
(XLSX)

**S6 File. Raw capillary flow profiles of 0.001% human plasma samples.**
(XLSX)

**S7 File. Raw capillary flow profiles of 0.001% human plasma samples with increased antibody-conjugated particles.**
(XLSX)

**S8 File. Processed flow velocity data of 0.001% human plasma samples.**
(XLSX)

## Acknowledgments

The authors thank Ms. Joy Abaidoo at Asthma and Airway Disease Research Center, The University of Arizona, who assisted in collecting blood samples from human volunteers.

## Author contributions

**Conceptualization:** Ashley K. Mathews, Allison J. Eby, Raymond K. Wong, Jeong-Yeol Yoon.

**Data curation:** Ashley K. Mathews, Allison J. Eby, Jocelyn Reynolds, Jeong-Yeol Yoon.

**Formal analysis:** Ashley K. Mathews, Allison J. Eby, Jocelyn Reynolds, Jeong-Yeol Yoon.

**Funding acquisition:** Jeong-Yeol Yoon.

**Investigation:** Ashley K. Mathews, Allison J. Eby, Jocelyn Reynolds.

**Methodology:** Ashley K. Mathews, Allison J. Eby, Jocelyn Reynolds, Jeong-Yeol Yoon.

**Project administration:** Raymond K. Wong, Jeong-Yeol Yoon.

**Resources:** Ashley K. Mathews, Allison J. Eby, Raymond K. Wong, Jeong-Yeol Yoon.

**Software:** Ashley K. Mathews, Jeong-Yeol Yoon.

**Supervision:** Raymond K. Wong, Jeong-Yeol Yoon.

**Validation:** Ashley K. Mathews, Allison J. Eby, Jeong-Yeol Yoon.

**Visualization:** Jeong-Yeol Yoon.

**Writing – original draft:** Ashley K. Mathews, Jeong-Yeol Yoon.

**Writing – review & editing:** Jeong-Yeol Yoon.

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
