## [Decision Letter · Decision Letter 0]

18 Sep 2025

Response to Reviewers
Revised Manuscript with Track Changes
Manuscript
**Journal Requirements:**
**Additional Editor Comments (if provided):**

- The sentence "Individuals who are predisposed to thrombotic events are likely to be ATIII-deficient, which could be hereditary or acquired over time," is confusing as there are indeed many other reasons one might be predisposed to thrombotic events outside of AT deficiency. - Reviewer #1 also points this out.

- Limitations section should make clear that this assay aims to replace quantitative AT assays but would not replace AT activity assays which are required to diagnose qualitative defects in AT function.

- The title should be revised as per Reviewer #2's comment given significant barriers to POC application at present; Reviewer #1 also points to current limitations in translating current assay to POC test.

Reviewer #1:

The authors detail a method for determining AT levels at the bedside over using traditional laboratory instrumentation. This manuscript describes proof of principle for determination of AT levels in a POC device.

1. The authors use PBS and human blood for these experiments and note differences between the two as expected as the authors discuss in the text. The addition of additional medications in patients that are undergoing procedures in hospitals present challenges for laboratory instrumentation when measuring coagulation parameters. Specifically, those that are in intensive care settings are often those that pose the greatest challenge. The authors should this as the next hurdle they will need to overcome to show that AT levels can be determined in the presence of yet more interference.

2. The statement in the introduction "Individuals who are predisposed to thrombotic events are likely to be ATIII-deficient" is misleading and inaccurate as there are many pathological reasons for a patient to be pro thrombotic and not just deficient in AT. Please revise this statement to reflect the general literature.

3. The section on ECMO seems out of place as AT do not be determined at the bedside, as this is done in a hospital laboratory. The rationale for rapid POC testing isn't presented in the paper for ECMO as daily monitoring of AT levels is common practice.

4. Given the loading requirements for the human whole blood were diluted how will this be achieved at the bedside? Dilutions are prone to errors and often impossible to perform outside of a laboratory as the tools for performing them aren't at the bedside.

5. Provide the details on how the dilutions were performed further to point 4.

6. IN the statistical analysis the R2 is good but the equation isn't discussed. Figure 4 shows a semi log transformation of the data to show a straight line which to this reviewer mean there is an exponential relationship between the time and the concentration of AT.

7. Could the authors comment about the use of such a device for other applications in coagulation where the clinical need is more urgent and the resources may not be available.

Reviewer #2:

General Assessment

This manuscript presents a technically innovative proof-of-concept for detecting Antithrombin III (ATIII) using smartphone-based flow velocity measurements on paper microfluidic chips. While the concept is novel and potentially impactful, the current study does not sufficiently support its translation into a point-of-care (POC) diagnostic tool.

Major Comments

1. POC Applicability and Validation

The authors claim that this assay could fill a gap in POC testing for ATIII. However, this claim is not well substantiated:

• The assay is not validated against the immunoassays currently used in clinical diagnostics, such as Immunoassays/ELISA or functional activity assays.

• There is no formal method comparison (e.g., Bland–Altman analysis, regression, or bias estimation) between the smartphone-based assay and the ELISA results.

• The clinical relevance and interchangeability of the smartphone-based flow assay with existing diagnostic standards remain unclear.

Recommendation: Include a direct method comparison using patient samples across the clinical range, with appropriate statistical analysis of agreement and bias.

2. Claim of Filling a Diagnostic Gap

The manuscript suggests that this method fills a critical gap in ATIII testing. However, this is not convincing:

• Portable immunoassay platforms (e.g., lateral flow, microfluidic ELISA) are already being developed for multiple analytes and are more directly translatable to POC settings.

• The proposed method requires precise dilution, controlled environmental conditions, and smartphone-based video analysis, which may not be feasible in real-world point-of-care (POC) environments typically operated by non-laboratorians.

Recommendation: Temper the claim and discuss alternative POC strategies. Justify why this method offers a unique advantage over portable immunoassay devices.

3. LoD Determination and Method Evaluation

The manuscript mentions sensitivity and detection limits but does not describe how Limit of Detection (LoD) was formally determined.

• There is no mention of CLSI guidelines (e.g., EP17-A2, EP05-A3, EP09-A3) or any standard framework for evaluating analytical performance.

• LoD appears to be inferred from calibration curves, but without statistical justification or replicate data.

Recommendation:

• Clearly define how LoD was calculated (e.g., blank + 1.645 × SD of low concentration sample).

• State whether CLSI or other recognized guidelines were consulted.

• Discuss precision, reproducibility, and analytical range in the context of clinical diagnostics.

Additionally, the manuscript should address how this method compares to in vitro diagnostic standards in terms of:

• Analytical sensitivity and specificity

• Bias and agreement

• Sample preparation complexity

• Turnaround time and operational feasibility

4. ELISA Kit Details (Line 219)

The manuscript references the use of a commercial ELISA kit to determine endogenous ATIII levels but lacks sufficient detail.

Recommendation: Provide:

• Full kit name, manufacturer, and catalog number (already partially listed as SimpleStep Human ATIII ELISA, Abcam).

• Lot number, calibration range, and sensitivity.

• Sample preparation and dilution protocols.

• Whether ELISA was used as a reference method for validation.

5. Device Translation and Usability

The manuscript does not address how the smartphone-based flow assay would be translated into a deployable device for clinical use.

Recommendation:

• Describe the envisioned device architecture (e.g., integrated chip-reader, automated sample loading).

• Discuss challenges in standardizing flow velocity measurements across different smartphones and environmental conditions.

• Address reproducibility and robustness in field settings.

Minor Comments

• Figures: Consider adding a schematic comparing the smartphone-based assay to a conventional immunoassay workflow.

• Terminology: Clarify units and dilution factors consistently (e.g., ng/mL vs mg/dL).

• Data Presentation: Include calibration curves and raw flow velocity data in supplementary materials.

Conclusion

This manuscript presents an interesting and technically sound proof-of-concept. However, its current form does not support the claim of clinical POC applicability. Additional validation, comparative analysis, and discussion of device translation are needed to strengthen the manuscript and justify its relevance to clinical diagnostics.

**Reviewers' Comments:**

**Comments to the Author**

1. Does this manuscript meet PLOS Digital Health’s publication criteria?

Reviewer #1: Yes

Reviewer #2: No

2. Has the statistical analysis been performed appropriately and rigorously?

Reviewer #1: Yes

Reviewer #2: No

3. Have the authors made all data underlying the findings in their manuscript fully available (please refer to the Data Availability Statement at the start of the manuscript PDF file)?

Reviewer #1: Yes

Reviewer #2: No

4. Is the manuscript presented in an intelligible fashion and written in standard English?

Reviewer #1: Yes

Reviewer #2: Yes

Reviewer #1: The authors detail a method for determining AT levels at the bedside over using traditional laboratory instrumentation. This manuscript describes proof of principle for determination of AT levels in a POC device.

1. The authors use PBS and human blood for these experiments and note differences between the two as expected as the authors discuss in the text. The addition of additional medications in patients that are undergoing procedures in hospitals present challenges for laboratory instrumentation when measuring coagulation parameters. Specifically, those that are in intensive care settings are often those that pose the greatest challenge. The authors should this as the next hurdle they will need to overcome to show that AT levels can be determined in the presence of yet more interference.

2. The statement in the introduction "Individuals who are predisposed to thrombotic events are likely to be ATIII-deficient" is misleading and inaccurate as there are many pathological reasons for a patient to be pro thrombotic and not just deficient in AT. Please revise this statement to reflect the general literature.

3. The section on ECMO seems out of place as AT do not be determined at the bedside, as this is done in a hospital laboratory. The rationale for rapid POC testing isn't presented in the paper for ECMO as daily monitoring of AT levels is common practice.

4. Given the loading requirements for the human whole blood were diluted how will this be achieved at the bedside? Dilutions are prone to errors and often impossible to perform outside of a laboratory as the tools for performing them aren't at the bedside.

5. Provide the details on how the dilutions were performed further to point 4.

6. IN the statistical analysis the R2 is good but the equation isn't discussed. Figure 4 shows a semi log transformation of the data to show a straight line which to this reviewer mean there is an exponential relationship between the time and the concentration of AT.

7. Could the authors comment about the use of such a device for other applications in coagulation where the clinical need is more urgent and the resources may not be available.

Reviewer #2: General Assessment

This manuscript presents a technically innovative proof-of-concept for detecting Antithrombin III (ATIII) using smartphone-based flow velocity measurements on paper microfluidic chips. While the concept is novel and potentially impactful, the current study does not sufficiently support its translation into a point-of-care (POC) diagnostic tool.

Please see attached reviewer's report for more details

**Do you want your identity to be public for this peer review?** For information about this choice, including consent withdrawal, please see our Privacy Policy

Reviewer #1: No

Reviewer #2: **Yes:** Sally Ezra

**Figure resubmission:**

**Reproducibility:** To enhance the reproducibility of your results, we recommend that authors of applicable studies deposit laboratory protocols in protocols.io, where a protocol can be assigned its own identifier (DOI) such that it can be cited independently in the future. Additionally, PLOS ONE offers an option to publish peer-reviewed clinical study protocols. Read more information on sharing protocols at https://plos.org/protocols?utm_medium=editorial-email&utm_source=authorletters&utm_campaign=protocols

---

## [Decision Letter · Decision Letter 1]

13 Jan 2026

Rapid Antithrombin Assay from Human Blood Plasma Utilizing Smartphone-Based Flow Observation on Paper Chips

PDIG-D-25-00557R1

Dear Prof. Yoon,

We are pleased to inform you that your manuscript 'Rapid Antithrombin Assay from Human Blood Plasma Utilizing Smartphone-Based Flow Observation on Paper Chips' has been provisionally accepted for publication in PLOS Digital Health.

Best regards,

Benjamin Chin-Yee, MD PhD FRCPC

Academic Editor

PLOS Digital Health

**Additional Editor Comments (if provided):**

**Reviewer Comments (if any, and for reference):**

Reviewer's Responses to Questions

**Comments to the Author**

Reviewer #1: All comments have been addressed

Reviewer #2: All comments have been addressed

publication criteria?

Reviewer #1: Yes

Reviewer #2: Yes

3. Has the statistical analysis been performed appropriately and rigorously?

Reviewer #1: Yes

Reviewer #2: N/A

4. Have the authors made all data underlying the findings in their manuscript fully available (please refer to the Data Availability Statement at the start of the manuscript PDF file)?

Reviewer #1: Yes

Reviewer #2: Yes

5. Is the manuscript presented in an intelligible fashion and written in standard English?

Reviewer #1: Yes

Reviewer #2: Yes

Reviewer #1: The authors have addressed all the comment brought forward by the reviewers and the manuscript now highlights the novel way to do anticoagulant testing with a smartphone

Reviewer #2: The authors have modified the manuscript and addressed all concerned

**Do you want your identity to be public for this peer review?** For information about this choice, including consent withdrawal, please see our Privacy Policy

Reviewer #1: **Yes:** Benjamin Hedley

Reviewer #2: **Yes:** Sally Ezra
